# PiD: Generalized AI-Generated Images Detection with Pixelwise Decomposition Residuals

**Xinghe Fu** [1]  **Zhiyuan Yan** [2]  **Zheng Yang** [1]  **Taiping Yao** [2]  **Yandan Zhao** [2]  **Shouhong Ding** [2]  **Xi Li** [1]

## Abstract

Fake images, created by recently advanced generative models, have become increasingly indistinguishable from real ones, making their detection crucial, urgent, and challenging. This paper introduces **PiD** (**Pi**xelwise **D**ecomposition Residuals), a novel detection method that focuses on residual signals within images. Generative models are designed to optimize high-level semantic content (principal components), often overlooking low-level signals (residual components). PiD leverages this observation by disentangling residual components from images, encouraging the model to uncover more underlying and general forgery clues independent of semantic content. Compared to prior approaches that rely on reconstruction techniques or high-frequency information, PiD is computationally efficient and does not rely on any generative models for reconstruction. Specifically, PiD operates at the pixel level, mapping the pixel vector to another color space (e.g., YUV) and then quantizing the vector. The pixel vector is mapped back to the RGB space and the quantization loss is taken as the residual for AIGC detection. Our experiment results are striking and highly surprising: PiD achieves 98% accuracy on the widely used GenImage benchmark, highlighting the effectiveness and generalization performance.

## 1. Introduction

The rapid advancement of generative models, such as Generative Adversarial Networks (GANs) and Diffusion Models, has revolutionized the field of image synthesis, enabling the creation of photorealistic images that are increasingly

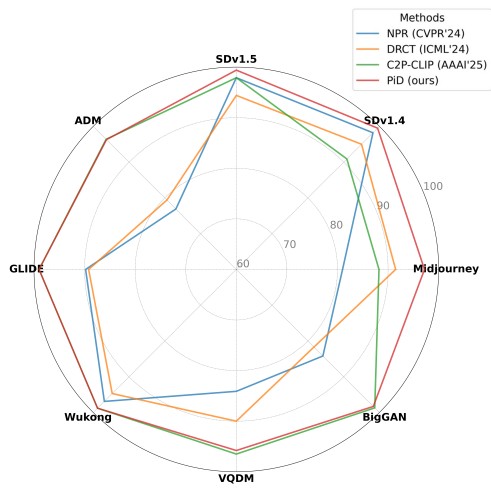

*Figure 1.* Cross-generators detection performance on the GenImage benchmark (Zhu et al., 2024) using the Accuracy metric. We compare our method (PiD) with existing SOTA detectors and demonstrate surprisingly superior results in generalization (98% Accuracy on average).

difficult to distinguish from real ones (Rombach et al., 2022; Zhan et al., 2023; Yan et al., 2025). Although these advanced technologies have opened up new possibilities in industries, they have also raised significant security concerns about the spread of disinformation. AI-generated fake images can be misused to misguide public opinion, fabricate evidence in forensic science, undermine trust in digital media, and more. Therefore, detecting such AI-generated images (AIGIs) has become a critical and urgent topic in both academia and industry.

One particularly effective detection strategy is using reconstruction. Reconstruction-based methods like DIRE (Wang et al., 2023), LARE[2] (Luo et al., 2024), and DRCT (Chen et al., 2024) have proven generalization performance for detecting AI-generated images by leveraging discrepancies between originals and their reconstructions. However, these approaches typically rely on heavy self-reconstruction generators (*e.g.,* DDIM in DIRE) to simulate how a generative model might rebuild an image, capitalizing on the idea that real images often fail to preserve low-level details during reconstruction. Their **high computational cost** is also

[1]College of Computer Science and Technology, Zhejiang University, Hangzhou, China [2]Youtu Lab, Tencent, Shanghai, China. Correspondence to: Xi Li <xilizju@zju.edu.cn>, Taiping Yao <taipingyao@tencent.com>.

*Proceedings of the 42nd International Conference on Machine Learning*, Vancouver, Canada. PMLR 267, 2025. Copyright 2025 by the author(s).

prohibitive for real-world applications, and more critically, detectors trained on these frameworks, with specific self-reconstruction generators involved, potentially being **overfitted to generator-specific artifacts**, resulting in a poor generalization in previously unseen generators (Ojha et al., 2023; Yan et al., 2023). For instance, diffusion-based self-construction generators are used in DIRE and DRCT, which struggle to generalize to GAN-generated images, as they learn patterns tied to diffusion priors rather than universal and general detection cues.

With the rapid development of vision-language models, recent works have explored them for detection by leveraging the pre-trained semantic knowledge within the vision-language models. CLIP (Radford et al., 2021), used in Uni-vFD (Ojha et al., 2023) and C2P-CLIP (Tan et al., 2024b), is used to identify inconsistencies in multimodal embeddings or perform linear classification based on semantic features. A more recent work, FatFormer (Liu et al., 2024), proposes a lightweight adapter and injects frequency information into the original CLIP model, enhancing its generalization performance. While promising, such methods risk obsolescence as generative models increasingly prioritize photorealism and semantic coherence. This underscores the enduring importance of low-level traces for robust detection.

Given the above concerns, the research question becomes: *how can we find a computationally simple, and at the same time, universal forgery artifact, without relying on generator-specific cues?*

To address this, in this paper, we propose a novel detection method, **PiD** (**Pi**xelwise **D**ecomposition residuals), which focuses on pixel-level residual signals within images. Unlike high-level semantic features, which are prioritized by generative models, residual components are typically overlooked during the image synthesis process. These residual signals, which include subtle pixel-level inconsistencies and quantization artifacts, provide a rich source of forgery clues that are independent of the semantic content of the image. By disentangling and analyzing these residual components, PiD can uncover underlying and general generative patterns that are difficult to discover from the original RGB image.

Specifically, our PiD operates at the pixel level, mapping pixel vectors to alternative linear transforms (e.g., YUV) and then quantizing them. The quantized vector is then mapped back to the RGB color space with the inverse transform. After that, the quantization loss, which represents the residual signal, is used as a key feature for AIGC detection. This approach is computationally efficient, as it avoids the need for complex reconstruction techniques or reliance on generative models. The superior generalization performance of the proposed method is shown in Fig. 1.

The key contributions of this work are three-fold:

- We propose a *computationally efficient, generator-free, yet highly effective method* based on pixelwise decomposition residuals, achieving superior generalization performance on widely used benchmarks.

- We propose a *new perspective to decompose the residual signal*: we first operate at the pixel level and map the pixel vectors to a *color space*, and then quantize the vector to produce the residual signals.

- We conduct extensive experiments on existing widely used benchmarks and demonstrate the *surprisingly high generalization performance* over other SOTAs.

## 2. Related Work

In this section, we briefly introduce the existing literature in the field of AIGI detection. Following Tan et al. (2024b); Yan et al. (2024c), we systematically categorize the field into two principal domains: Face Forgery Detection and AIGC Detection for discussion.

### 2.1. Face Forgery Detection

Face forgery (classical deepfake) detection has been a prominent area of research due to the quick rise of face-swapping and face-reenactment. The domain of face forgery detection has seen substantial advancements, with numerous studies concentrating on the exploitation of spatial or frequency information derived from images. Face forgery detection methods can be broadly categorized into **high-level** and **low-level** approaches. High-level methods focus on biological or behavioral inconsistencies, such as irregular eye-blinking patterns (Li et al., 2018) or lip-sync artifacts (Haliassos et al., 2021), to exploit unnatural facial dynamics in deepfakes. In contrast, most works target low-level cues, including spatial artifacts (*e.g.,* using Xception (Chollet, 2017) to detect texture anomalies (Rossler et al., 2019b)) or frequency artifacts (Luo et al., 2021; Qian et al., 2020; Masi et al., 2020; Woo et al., 2022) caused by forgery generation processes. To enhance generalization, recent strategies diversify training data via adversarial perturbations (Chen et al., 2022), challenging fake data synthesis (Li et al., 2020; Shiohara et al., 2022; Cheng et al., 2024), or disentanglement-based representation learning (UCF (Yan et al., 2023) and UDD (Fu et al., 2025)), or latent space simulations (LSDA (Yan et al., 2024a)) to capture invariant forgery features across manipulation techniques.

### 2.2. AIGC Detection

The rapid evolution of generative technologies has expanded synthetic content beyond facial forgeries to diverse scenes, driving research toward AIGC detection: a task more complex than traditional face forgery detection due to its broader

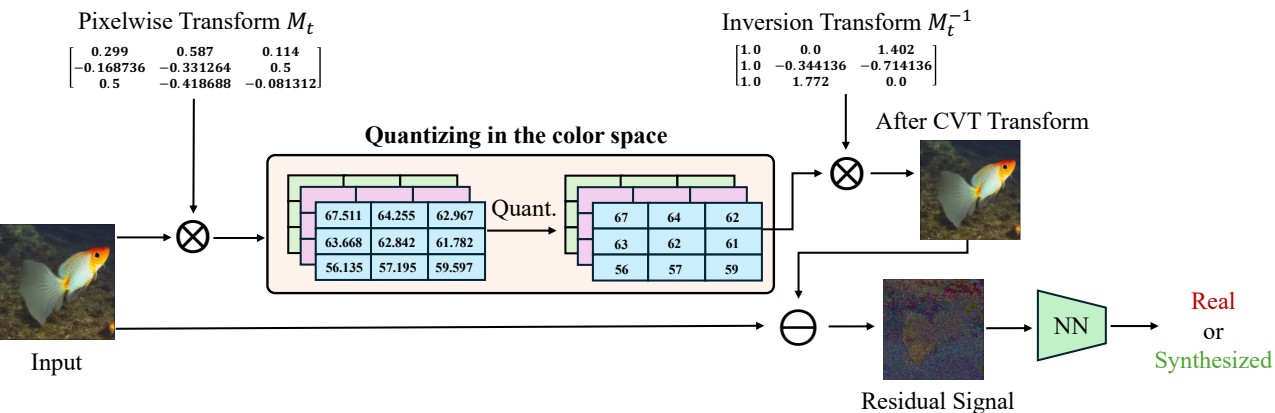

*Figure 2.* The pipeline of the proposed PiD method. A pixelwise transformation matrix $M_t$ is applied to the input RGB image, and a quantization operation is appended to the projected color space. The quantized image is then mapped back to the RGB space with $M_t^{-1}$. The residual information is decomposed by subtracting the quantized image from the original image. The residual signal is sent to the detection network like ResNet (He et al., 2016) to distinguish real and synthetic images.

forgery types and heightened demands for generalization. Current methodologies address this challenge through two complementary lenses: **low-level artifact analysis** and **high-level semantic cues**. Low-level approaches target subtle statistical irregularities induced during content generation: CNN-Spot (Wang et al., 2020) employs data augmentation to improve generalization, while BiHPF (Jeong et al., 2022) amplifies artifacts via dual high-pass filters, LGrad (Tan et al., 2023a) extracts gradient-based patterns, NPR (Tan et al., 2024d) models neighboring pixel relationships, and random-mapping features (Tan et al., 2024a) expose forgery-specific distortions. High-level methods, conversely, leverage semantic inconsistencies in synthetic content: UnivFD (Ojha et al., 2023) adopts CLIP embeddings for zero-shot detection, FatFormer (Liu et al., 2024) integrates frequency analysis with vision-language alignment from CLIP, and LASTED (Wu et al., 2023) exploits text-guided contrastive learning to identify mismatches between visual and textual semantics. By synergizing low-level artifact detection with high-level semantic reasoning, these strategies collectively fortify defenses against the growing sophistication of AIGC threats.

## 3. Method

### 3.1. Problem Setup

The detection of AI-generated images (AIGI) is formulated as a binary classification task. Given an input image $x \in \mathbb{R}^{H \times W \times 3}$, a neural network $f_\theta(\cdot)$ predicts the probability $p = f_\theta(x) \in [0, 1]$ that $x$ is synthesized by generative models. The training objective minimizes the cross-entropy loss as follows:

$$\mathcal{L}_{\text{CE}} = -\frac{1}{N} \sum_{i=1}^{N} \left[ y_i \log p_i + (1 - y_i) \log(1 - p_i) \right], \quad (1)$$

where $y_i \in \{0, 1\}$ is the ground-truth label, and $N$ is the batch size. However, directly taking the original image $x$ as input in AIGI detection is usually challenging due to the high quality of synthesized images. It requires a powerful pre-trained network as the detector to achieve high detection accuracy. As an alternative, it is crucial to find an image representation that is difficult for the current generative models to fit.

Modern generative models (e.g., diffusion models (Ho et al., 2020)) optimize the mean squared error (MSE) during training:

$$\mathcal{L}_{\text{MSE}} = \mathbb{E}[(x - G(x, \epsilon))^2], \quad (2)$$

where $G(\cdot)$ is the generator and $\epsilon$ denotes noise. The MSE loss decomposes into:

$$\mathcal{L}_{\text{MSE}} = \underbrace{(\mathbb{E}[G(x, \epsilon)] - \mathbb{E}[x])^2}_{\text{Bias}^2}$$
$$+ \underbrace{Var[G(x, \epsilon)]}_{\text{Variance}} + \underbrace{\mathbb{E}[(x - \mathbb{E}[x])^2]}_{\text{Noise}}. \quad (3)$$

Critically, the *noise* term corresponds to data-specific artifacts that remain irreducible even for high-fidelity synthetic images. To exploit this in AIGI detection, a general approach is to extract **noise-aware residual representation** $R(x)$ from an image $x$, which disentangles generative noise patterns from semantic content, defined as:

$$R(x) = x - \Phi(x), \quad (4)$$

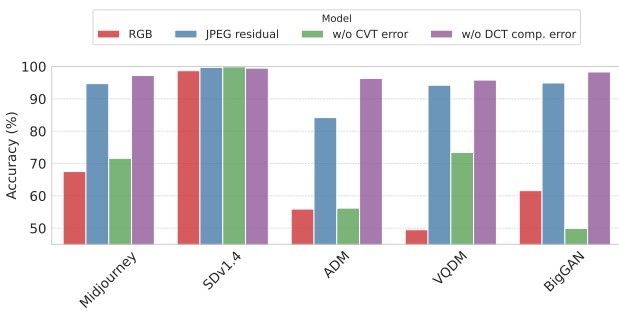

*Figure 3.* Comparison of RGB input and different image residual inputs. Models are trained on the SDv1.4 training set. The color conversion (CVT) error in the residual contributes most to the generalization performance.

where $R(x)$ represents the noise part of $x$, $\Phi(x)$ represents the intrinsic value of image $x$. Prior works (Wang et al., 2023; Tan et al., 2024c;d; Ricker et al., 2024) use different methods to estimate the function $\Phi(x)$ (e.g., low-frequency filters or the output of generation models), and the residual information is extracted as follows.

- **Frequency-based residuals:** Methods like FreqNet (Tan et al., 2024c) extract high-frequency components via:

$$R_{\text{freq}}(x) = \mathcal{F}^{-1}\left(\mathcal{F}(x) \odot M_{\text{high}}\right), \qquad (5)$$

where $F(\cdot)$ is the Fourier transform and $M_{high}$ is a high-pass filter. This is equal to subtract the low-frequency components from the image and obtain the residual information. However, the noise pattern cannot be fully captured by specific frequency bands.

- **Reconstruction-based residuals:** Methods like DIRE (Wang et al., 2023) compute $R_{dire}(x) = x - \hat{x}$, where $\hat{x} = D(G(x))$ is the output of a pre-trained generative model (e.g., diffusion models). However, the output inherits biases from the generator $G(\cdot)$, limiting generalization to unseen generators.

### 3.2. Image Compression Residuals

We aim to find a more generalizable method to extract the residual representation $R(x)$ from $x$ for AIGI detection. To achieve this purpose, we have to consider how to approximate the intrinsic values of an image $x$. Notice that the compression algorithm, like JPEG, filters the noise information and maintains the visual quality of images. We first explore the residual information between the original image and the compressed image.

Compression algorithms (e.g., JPEG) also have an encoder-decoder structure like many generative models. For the

---

**Algorithm 1** PiD Training Pipeline

**Input:** dataset $D$, neural network $f_\theta(\cdot)$, transformation matrix $M_t$, quantization function $Q(\cdot)$
Initialize the parameters $\theta$ for $f_\theta(\cdot)$.
**for** $epoch = 1$ **to** $nepochs$ **do**
    **for** $i = 1$ **to** $niters$ **do**
        Fetch batch $(x_i, y_i)$ from $D$
        $x_i' = x_i \times M_t$
        $x_i' = Q(x_i')$ {Rounding or truncation}
        $x_i' = x_i' \times M_t^{-1}$
        $R_x = x_i - x_i'$ {Compute residual representation}
        $p_i = f_\theta(R_x)$ {Neural network output}
        Compute $L_{CE}(p_i, y_i)$
        Update $\theta$ with $\nabla_\theta L_{CE}$
    **end for**
**end for**
Return $f_\theta(\cdot)$

---

compression-based residual representation $R_{comp}(x)$, suppose that $D(\cdot)$ and $E(\cdot)$ are the encoder and decoder in the compression, the definition is as follows,

$$R_{comp}(x) = x - D(E(x)). \qquad (6)$$

Taking the JPEG compression algorithm as an example, we explore and verify the effectiveness of the compression residual $R_{comp}(x)$ and the essential operations. The residual information $R_{comp}(x)$ mainly includes two stages of quantization loss. One is the quantization loss during color space conversion, and the other is the quantization compression loss of the blockwise DCT (discrete cosine transform) frequency components.

Experiments have found that the compression-based residual representation can effectively achieve generalization on different generative models, as shown in Figure 3. However, if only the frequency-domain compression loss is included in the residual, the generalization effect is limited. This indicates that pixel-wise compression quantization of the image is the key to capturing the noise information of the image. The results reveal that a simple pixelwise operation can effectively extract the noise patterns from the image.

### 3.3. Pixelwise Decomposition in AIGI Detection

We propose a pixelwise decomposition operation inspired by the color conversion. Suppose that an invertible transformation matrix $M_t \in \mathbb{R}^{3\times3}$ (e.g., YUV conversion matrix as shown in Figure 2) and a quantization function (like rounding or truncation function) is applied to a pixel vector, the pixel vector can be decomposed into a quantized vector and residual vector. The formal definition of the pixelwise decomposition residual is given as follows.

**Definition 3.1.** *With a invertible matrix $M_t \in \mathbb{R}^{3\times3}$, its inversion matrix $M_t^{-1} \in \mathbb{R}^{3\times3}$ and a quantization function $Q(\cdot)$, given an image $x \in [0, 255]^{3 \times H \times W}$, the pixelwise decomposition residual is defined as*

$$R_{pid}(x) := x - Q(x \times_1 M_t) \times_1 M_t^{-1}. \qquad (7)$$

$\times_1$ is the operation of the mode-1 tensor-matrix product.

Our training pipeline takes $R_{pid}(x)$ as the input and detects synthetic images in the noise rather than the original RGB space. The detection model becomes $f_\theta(R_{pid}(x))$, and the cross-entropy loss $L_{CE}$ is still the supervision during training. The training pipeline is shown in Algorithm 1.

The PiD residual $R_{pid}(x)$ separates the noise information from the visual content of image $x$, since the transformation has almost no effect on the visual quality (where $\|R_{pid}(x)\| << \|x\|$). Moreover, as the decomposition is applied to pixels independently, the noise information is well preserved. This residual representation is not limited to specific global or local frequency bands (as in $R_{freq}(x)$) or affected by generative models' bias (as in $R_{dire}(x)$). In the experiment section, we will show that PiD is generalizable over different generative models.

## 4. Experiment

### 4.1. Setup

**Training datasets.** We consider different training settings following ForenSynths (Wang et al., 2020) and GenImage (Zhu et al., 2024) with ProGAN (Karras et al., 2018) or SDv1.4 (Rombach et al., 2022) as the generative model. The real images in ForenSynths and GenImage are from LSUN (Yu et al., 2015) and ImageNet (Russakovsky et al., 2015) dataset. There are 20 semantic classes of images in ForenSynths and we only use 4 of them (i.e., car, cat, chair, horse) during training to keep in line with previous works (Tan et al., 2024c;d; Liu et al., 2024).

**Test datasets.** To evaluate the generalization performance of different approaches in real-world scenarios, we test the models on 3 widely used datasets with 26 generative models. Synthetic images in these datases are generated by diverse GANs and DMs and real images are from different sources.

- **UniversalFakeDetect dateset** (Ojha et al., 2023). This dataset is composed of the test set from ForenSynths (Wang et al., 2020) and some additional synthetic images generated by DMs. 11 generative models (7 GANs and 4 DMs) are included during the test. These models are ProGAN (Karras et al., 2018), CycleGAN (Zhu et al., 2017), BigGAN (Brock et al., 2018), StyleGAN (Karras et al., 2019), GauGAN (Park et al., 2019), StarGAN (Choi et al., 2018), Deepfakes (Rossler et al., 2019b), Guided (Dhariwal et al.,

2021), LDM (Rombach et al., 2022), GLIDE (Nichol et al., 2021), and DALLE (Ramesh et al., 2022). Different sampling strategies are applied for some of the DMs (i.e., LDM (Rombach et al., 2022) and GLIDE (Nichol et al., 2021)). Multi-source real images (e.g., LSUN (Yu et al., 2015), ImageNet (Russakovsky et al., 2015), FaceForensics++ (Rossler et al., 2019a), and LAION (Schuhmann et al., 2021)) are collected to comprehensively evaluate the performance of detectors.

- **GenImage dataset** (Zhu et al., 2024). This dataset is composed of synthetic images from advanced generative models (7 DMs and a GAN method). These models are Midjourney (Mid, 2022), SDv1.4 (Rombach et al., 2022), SDv1.5 (Rombach et al., 2022), ADM (Dhariwal et al., 2021), GLIDE (Nichol et al., 2021), Wukong (Wuk, 2022.5), VQDM (Egiazarian et al., 2024), and BigGAN (Brock et al., 2018). Compared with other datasets, the visual quality of synthetic images in GenImage is better and more challenging for many of the previous detection methods. The varying resolutions in the GenImage dataset also pose higher demands for the generalization ability of detection models.

- **Self-Synthesis GAN dataset** (Tan et al., 2024c). 9 advanced GAN techniques are included in this dataset to enrich the existing test scene (e.g., ForenSynths (Wang et al., 2020)). These models are AttGAN (He et al., 2019), BEGAN (Berthelot et al., 2017), CramerGAN (Bellemare et al., 2017), InfoMaxGAN (Lee et al., 2021), MMDGAN (Li et al., 2017), RelGAN (Nie et al., 2019), S3GAN (Lučić et al., 2019), SNGAN (Miyato et al., 2018), and S3GAN (Lučić et al., 2019).This dataset is challenging and provides a robust test scene for AIGI detection models trained on ForenSynths.

**Implementation details.** We implement the detector network with a simple customized ResNet architecture (Tan et al., 2024d; Li et al., 2024). The transformation matrix $M_t$ used in PiD is the matrix $M_{YUV}$ that maps the pixel value from RGB space to YUV color space by default. Rounding (*round*) or truncation (*floor*) functions are utilized as the quantization function $Q(\cdot)$ in the implementation. An additional rounding operation is appended after the inversion transformation in the RGB space by default. We train the detector network for 50 epochs with batch size 64. The network is optimized with an SGD optimizer with a learning rate of 0.001.

**Metrics.** We follow existing works (Ojha et al., 2023; Liu et al., 2024; Tan et al., 2024d) for benchmarking and report both average precision (AP) and classification accuracy (Acc). For Acc, we set the classification threshold for each dataset to 0.5 to ensure a fair comparison.

*Table 1.* **Cross-model accuracy (%) performance on the UniversalFakeDetect Dataset.** All models are trained on ForenSynths (ProGAN) under the 4-class setting. **Bold** and underline represent the best and second-best performance.

| Methods | Venue | GAN | | | | | | Deep fakes | Guided | LDM | | | GLIDE | | | DALLE | mAcc |
|---|---|---|---|---|---|---|---|---|---|---|---|---|---|---|---|---|---|
| | | Pro-GAN | Cycle-GAN | Big-GAN | Style-GAN | Gau-GAN | Star-GAN | | | 200 steps | 200 w/cfg | 100 steps | 100 27 | 50 27 | 100 10 | | |
| CNN-Spot | CVPR2020 | 99.99 | 85.20 | 70.20 | 85.7 | 78.95 | 91.7 | 53.47 | 60.07 | 54.03 | 54.96 | 54.14 | 60.78 | 63.8 | 65.66 | 55.58 | 68.95 |
| Patchfor | ECCV2020 | 75.03 | 68.97 | 68.47 | 79.16 | 64.23 | 63.94 | 75.54 | 67.41 | 76.5 | 76.1 | 75.77 | 74.81 | 73.28 | 68.52 | 67.91 | 71.71 |
| Co-occurence | Elect. Imag. | 97.70 | 63.15 | 53.75 | 92.50 | 51.1 | 54.7 | 57.1 | 60.50 | 70.7 | 70.55 | 71.00 | 70.25 | 69.60 | 69.90 | 67.55 | 68.00 |
| Freq-spec | WIFS2019 | 49.90 | **99.90** | 50.50 | 49.90 | 50.30 | 99.70 | 50.10 | 50.90 | 50.40 | 50.40 | 50.30 | 51.70 | 51.40 | 50.40 | 50.00 | 57.05 |
| F3Net | ECCV2020 | 99.38 | 76.38 | 65.33 | 92.56 | 58.10 | **100.0** | 63.48 | 69.20 | 68.15 | 75.35 | 68.80 | 81.65 | 83.25 | 83.05 | 66.30 | 76.73 |
| UnivFD | CVPR2023 | **100.0** | 98.50 | 94.50 | 82.00 | 99.50 | 97.00 | 66.60 | 70.03 | 94.19 | 73.76 | 94.36 | 79.07 | 79.85 | 78.14 | 86.78 | 86.29 |
| LGrad | CVPR2023 | 99.84 | 85.39 | 82.88 | 94.83 | 72.45 | 99.62 | 58.00 | 77.50 | 94.20 | 95.85 | 94.80 | 87.40 | 90.70 | 89.55 | 88.35 | 87.42 |
| FreqNet | AAAI2024 | 97.90 | 95.84 | 90.45 | 97.55 | 90.24 | 93.41 | **97.40** | 86.70 | 84.55 | 99.58 | 65.56 | 85.69 | 97.40 | 88.15 | 59.06 | 88.63 |
| NPR | CVPR2024 | 99.84 | 95.00 | 87.55 | 96.23 | 86.57 | 99.75 | 76.89 | 84.55 | 97.65 | 98.00 | 98.20 | 96.25 | 97.15 | 97.35 | 87.15 | 93.21 |
| FatFormer | CVPR2024 | 99.89 | 99.32 | **99.50** | 97.15 | 99.41 | 99.75 | 93.23 | 76.00 | 98.60 | 94.90 | 98.65 | 94.35 | 94.65 | 94.20 | **98.75** | 95.89 |
| C2P-CLIP | AAAI2025 | 99.98 | 97.31 | 99.12 | 96.44 | 99.17 | 99.60 | 93.77 | 69.10 | 99.25 | 97.25 | 99.30 | 95.25 | 95.25 | 96.10 | 98.55 | 95.70 |
| PiD ($M_{YUV}$, round) | | 99.81 | 99.76 | 93.79 | **99.85** | **100.0** | 91.34 | 95.45 | 82.60 | **99.95** | **99.95** | **99.45** | **99.20** | **99.30** | **99.30** | 83.55 | **96.22** |
| PiD ($M_{YUV}$, floor) | | **100.0** | 98.57 | 86.06 | 97.54 | 99.95 | 85.30 | 95.06 | **96.60** | 98.65 | 98.55 | 98.10 | 98.60 | 98.45 | 98.55 | 87.35 | 95.82 |

*Table 2.* **Cross-model Average Precision (AP) Performance on the UniversalFakeDetect Dataset.** **Bold** and underline represent the best and second-best performance.

| Methods | Venue | GAN | | | | | | Deep fakes | Guided | LDM | | | GLIDE | | | DALLE | mAP |
|---|---|---|---|---|---|---|---|---|---|---|---|---|---|---|---|---|---|
| | | Pro-GAN | Cycle-GAN | Big-GAN | Style-GAN | Gau-GAN | Star-GAN | | | 200 steps | 200 w/cfg | 100 steps | 100 27 | 50 27 | 100 10 | | |
| CNN-Spot | CVPR2020 | **100.0** | 93.47 | 84.5 | 99.54 | 89.49 | 98.15 | 89.02 | 73.72 | 70.62 | 71.0 | 70.54 | 80.65 | 84.91 | 82.07 | 70.59 | 83.88 |
| Patchfor | ECCV2020 | 80.88 | 72.84 | 71.66 | 85.75 | 65.99 | 69.25 | 76.55 | 75.03 | 87.1 | 86.72 | 86.4 | 85.37 | 83.73 | 78.38 | 75.67 | 78.75 |
| Co-occurence | Elect. Imag. | 99.74 | 80.95 | 50.61 | 98.63 | 53.11 | 67.99 | 59.14 | 70.20 | 91.21 | 89.02 | 92.39 | 89.32 | 88.35 | 82.79 | 80.96 | 79.63 |
| Freq-spec | WIFS2019 | 55.39 | **100.0** | 75.08 | 55.11 | 66.08 | **100.0** | 45.18 | 57.72 | 77.72 | 77.25 | 76.47 | 68.58 | 64.58 | 61.92 | 67.77 | 69.92 |
| F3Net | ECCV2020 | 99.96 | 84.32 | 69.90 | 99.72 | 56.71 | **100.0** | 78.82 | 70.53 | 73.76 | 81.66 | 74.62 | 89.81 | 91.04 | 90.86 | 71.84 | 82.24 |
| UnivFD | CVPR2023 | **100.0** | 99.46 | 99.59 | 97.24 | 99.98 | 99.60 | 82.45 | 87.77 | 99.14 | 92.15 | 99.17 | 94.74 | 95.34 | 94.57 | 97.15 | 95.89 |
| LGrad | CVPR2023 | **100.0** | 93.98 | 90.69 | 99.86 | 79.36 | 99.98 | 67.91 | 87.06 | 99.03 | 99.16 | 99.18 | 93.23 | 95.10 | 94.93 | 97.23 | 93.11 |
| FreqNet | AAAI2024 | 99.92 | 99.63 | 96.05 | 99.89 | 99.71 | 98.63 | **99.92** | 96.27 | 96.06 | **100.0** | 62.34 | **99.87** | 99.89 | 96.39 | 77.78 | 94.81 |
| NPR | CVPR2024 | **100.0** | 99.53 | 94.53 | 99.94 | 88.82 | **100.0** | 84.41 | 98.26 | 99.92 | 99.91 | 99.92 | **99.87** | 99.89 | 99.92 | 99.26 | 97.61 |
| FatFormer | CVPR2024 | **100.0** | **100.0** | **99.98** | 99.75 | **100.0** | **100.0** | 97.99 | 91.99 | 99.81 | 99.09 | 99.87 | 99.13 | 99.41 | 99.20 | 99.82 | **99.07** |
| C2P-CLIP | AAAI2025 | **100.0** | **100.0** | 99.96 | 99.50 | **100.0** | **100.0** | 98.59 | 94.13 | 99.99 | 99.83 | **99.98** | 99.72 | 99.79 | 99.83 | **99.91** | 98.66 |
| PiD ($M_{YUV}$, round) | | 99.99 | **100.0** | 94.98 | **99.95** | **100.0** | 96.93 | 97.58 | 93.28 | **100.0** | **100.0** | 99.75 | 99.77 | 99.85 | **99.96** | 91.73 | 98.24 |
| PiD ($M_{YUV}$, floor) | | 99.97 | 99.98 | 92.01 | 99.04 | **100.0** | 90.75 | 97.58 | **99.18** | 99.89 | 99.82 | 99.52 | **99.87** | **99.91** | 99.88 | 94.26 | 98.11 |

## 4.2. Cross-Model Evaluation

**Evaluation on UniversalFakeDetect.** The test results on UniversalFakeDetect are illustrated in Table 1 and Table 2, and the accuracy and the AP on each subset is reported. All the detection models are trained on the training set of ForenSynths with a single generative model ProGAN (Karras et al., 2018). Cross-model test is challenging for early detection methods like CNN-Spot (Wang et al., 2020), Patch-for (Chai et al., 2020), Co-occurence (Nataraj et al., 2019) and Freq-spec (Zhang et al., 2019). The average accuracy over 15 test sets is below 71.71%. Methods that focus on low-level representation like specific frequency components or image-level network gradients, i.e., F3Net (Qian et al., 2020), LGrad (Tan et al., 2023b), FreqNet (Tan et al., 2024c), and NPR (Tan et al., 2024d), achieve better gner-alization performance. NPR achieves an average accuracy of 93.21%, which leverages the local high-frequency information as input of the detector. Currently, some methods, i.e., UnivFD (Ojha et al., 2023), FatFormer (Liu et al., 2024), and C2P-CLIP (Tan et al., 2024b), find that utilizing pre-trained large visual backbone network (e.g., CLIP ViT-L (Radford et al., 2021)) can achieve remarkable performance in AIGI detection. Our method PiD shows com-putational efficiency with simple pixelwise transformation and lightweight network structure and surpass previous representation-based and pretraining-based methods. The proposed method achieve a new state-of-the-art result with an average accuracy of 96.22%, compared with NPR with 93.21% and FatFormer with 95.89%. The mAP of 98.24% is also competitive compared with pretraining-based methods like FatFormer and C2P-CLIP.

**Evaluation on GenImage.** To verify the generalization capability of the proposed method, we also test the models on another benchmark GenImage with advanced diffusion models as the training and test set. All the detection models are trained on the training set with SDv1.4 as the generative model. Some reconstruction-based methods use diffusion reconstruction to extract residual information in the RGB space, e.g., DIRE (Wang et al., 2023), or in the latent space, e.g., LARE[2] (Luo et al., 2024) and achieves great detection performance on diffusion models. However, the model bias of generative models may limit the generalization capability of the reconstruction-based method. As shown in Table 3, our method also achieves a new state-of-the-art result with an average accuracy of 98.0% over 8 generative models. PiD surpass the reconstruction-based method LARE[2] by 11.8%,

*Table 3.* **Cross-model accuracy (Acc) performance on the Genimage Dataset.** The SDv1.4 is employed as the training set following (Zhu et al., 2024). The results of ResNet-50, DeiT-S, Swin-T, CNNSpot, Spec, F3Net, and GramNet are from GenImage (Zhu et al., 2024). **Bold** and underline represent the best and second-best performance.

| Method | Venue | Midjourney | SDv1.4 | SDv1.5 | ADM | GLIDE | Wukong | VQDM | BigGAN | mAcc |
|---|---|---|---|---|---|---|---|---|---|---|
| ResNet-50 (He et al., 2016) | CVPR2016 | 54.9 | 99.9 | 99.7 | 53.5 | 61.9 | 98.2 | 56.6 | 52.0 | 72.1 |
| DeiT-S (Touvron et al., 2021) | ICML2021 | 55.6 | 99.9 | 99.8 | 49.8 | 58.1 | 98.9 | 56.9 | 53.5 | 71.6 |
| Swin-T (Liu et al., 2021) | ICCV2021 | 62.1 | 99.9 | 99.8 | 49.8 | 67.6 | 99.1 | 62.3 | 57.6 | 74.8 |
| CNNSpot (Wang et al., 2020) | CVPR2020 | 52.8 | 96.3 | 95.9 | 50.1 | 39.8 | 78.6 | 53.4 | 46.8 | 64.2 |
| Spec (Zhang et al., 2019) | WIFS2019 | 52.0 | 99.4 | 99.2 | 49.7 | 49.8 | 94.8 | 55.6 | 49.8 | 68.8 |
| F3Net (Qian et al., 2020) | ECCV2020 | 50.1 | 99.9 | **99.9** | 49.9 | 50.0 | 99.9 | 49.9 | 49.9 | 68.7 |
| GramNet (Liu et al., 2020) | CVPR2020 | 54.2 | 99.2 | 99.1 | 50.3 | 54.6 | 98.9 | 50.8 | 51.7 | 69.9 |
| UnivFD (Ojha et al., 2023) | CVPR2023 | 93.9 | 96.4 | 96.2 | 71.9 | 85.4 | 94.3 | 81.6 | 90.5 | 88.8 |
| DIRE (Wang et al., 2023) | ICCV2023 | 50.4 | **100.0** | 99.9 | 52.3 | 67.2 | **100.0** | 50.1 | 50.0 | 71.2 |
| FreqNet (Tan et al., 2024c) | AAAI2024 | 89.6 | 98.8 | 98.6 | 66.8 | 86.5 | 97.3 | 75.8 | 81.4 | 86.8 |
| NPR (Tan et al., 2024d) | CVPR2024 | 81.0 | 98.2 | 97.9 | 76.9 | 89.8 | 96.9 | 84.1 | 84.2 | 88.6 |
| FatFormer (Liu et al., 2024) | CVPR2024 | 92.7 | **100.0** | 99.9 | 75.9 | 88.0 | 99.9 | **98.8** | 55.8 | 88.9 |
| LARE² (Luo et al., 2024) | CVPR2024 | 74.0 | **100.0** | 99.9 | 61.7 | 88.5 | **100.0** | 97.2 | 68.7 | 86.2 |
| DRCT (Chen et al., 2024) | ICML2024 | 91.5 | 95.0 | 94.4 | 79.4 | 89.2 | 94.7 | 90.0 | 81.7 | 89.5 |
| Effort (Yan et al., 2024b) | ICML2025 | 82.4 | 99.8 | 99.8 | 78.7 | 93.3 | 97.4 | 91.7 | 77.6 | 91.1 |
| C2P-CLIP (Tan et al., 2024b) | AAAI2025 | 88.2 | 90.9 | 97.9 | **96.4** | **99.0** | 98.8 | 96.5 | **98.7** | 95.8 |
| PiD ($M_{YUV}$, round) | | **97.2** | 99.5 | 99.4 | 96.3 | **99.0** | 98.8 | 95.8 | 98.3 | **98.0** |
| PiD ($M_{YUV}$, floor) | | 95.5 | 99.6 | 99.5 | 96.0 | 98.5 | 99.3 | 97.8 | 96.2 | 97.8 |

*Table 4.* **Cross-GAN-sources evaluation on the Self-Synthesis 9 GANs dataset.** Acc and AP are reported for comparison. **Bold** and underline represent the best and second-best performance.

| Method | AttGAN Acc | AttGAN AP | BEGAN Acc | BEGAN AP | CramerGAN Acc | CramerGAN AP | InfoMaxGAN Acc | InfoMaxGAN AP | MMDGAN Acc | MMDGAN AP | RelGAN Acc | RelGAN AP | S3GAN Acc | S3GAN AP | SNGAN Acc | SNGAN AP | STGAN Acc | STGAN AP | Mean Acc | Mean AP |
|---|---|---|---|---|---|---|---|---|---|---|---|---|---|---|---|---|---|---|---|---|
| CNNDetection (Wang et al., 2020) | 51.1 | 83.7 | 50.2 | 44.9 | 81.5 | 97.5 | 71.1 | 94.7 | 72.9 | 94.4 | 53.3 | 82.1 | 55.2 | 66.1 | 62.7 | 90.4 | 63.0 | 92.7 | 62.3 | 82.9 |
| Frank (Frank et al., 2020) | 65.0 | 74.4 | 39.4 | 39.9 | 31.0 | 36.0 | 41.1 | 41.0 | 38.4 | 40.5 | 69.2 | 96.2 | 69.7 | 81.9 | 48.4 | 47.9 | 25.4 | 34.0 | 47.5 | 54.7 |
| Durall (Durall et al., 2020) | 39.9 | 38.2 | 48.2 | 30.9 | 60.9 | 67.2 | 50.1 | 51.7 | 59.5 | 65.5 | 80.0 | 88.2 | 87.3 | 97.0 | 54.8 | 58.9 | 62.1 | 72.5 | 60.3 | 63.3 |
| Patchfor (Chai et al., 2020) | 68.0 | 92.9 | 97.1 | **100.0** | 97.8 | 99.9 | 93.6 | 98.2 | 97.9 | **100.0** | 99.6 | **100.0** | 66.8 | 68.1 | 97.6 | 99.8 | 92.7 | 99.8 | 90.1 | 95.4 |
| F3Net (Qian et al., 2020) | 85.2 | 94.8 | 87.1 | 97.5 | 89.5 | 99.8 | 67.1 | 83.1 | 73.7 | 99.6 | 98.8 | **100.0** | 65.4 | 70.0 | 51.6 | 93.6 | 60.3 | 99.9 | 75.4 | 93.1 |
| SelfBlend (Shiohara et al., 2022) | 63.1 | 66.1 | 56.4 | 59.0 | 75.1 | 82.4 | 79.0 | 82.5 | 68.6 | 74.0 | 73.6 | 77.8 | 53.2 | 53.9 | 61.6 | 65.0 | 61.2 | 66.7 | 65.8 | 69.7 |
| GANDetection (Mandelli et al., 2022) | 57.4 | 75.1 | 67.9 | **100.0** | 67.8 | 99.7 | 67.6 | 92.4 | 67.7 | 99.3 | 60.9 | 86.2 | 69.6 | 83.5 | 66.7 | 90.6 | 69.6 | 97.2 | 66.1 | 91.6 |
| LGrad (Tan et al., 2023b) | 68.6 | 93.8 | 69.9 | 89.2 | 50.3 | 54.0 | 71.1 | 82.0 | 57.5 | 67.3 | 89.1 | 99.1 | 78.5 | 86.0 | 78.0 | 87.4 | 54.8 | 68.0 | 68.6 | 80.8 |
| UnivFD (Ojha et al., 2023) | 78.5 | 98.3 | 72.0 | 98.9 | 77.6 | 99.8 | 77.6 | 98.9 | 77.6 | 99.7 | 78.2 | 98.7 | 85.2 | **98.1** | 77.6 | 98.7 | 74.2 | 97.8 | 77.6 | 98.8 |
| NPR (Tan et al., 2024d) | 83.0 | 96.2 | 99.0 | 99.8 | 98.7 | 99.0 | 94.5 | 98.3 | 98.6 | 99.0 | 99.6 | **100.0** | 79.0 | 80.0 | 88.8 | 97.4 | 98.0 | **100.0** | 93.2 | 96.6 |
| PiD ($M_{YUV}$, round) | **100.0** | **100.0** | 99.9 | **100.0** | 95.4 | 99.7 | 95.4 | 99.8 | 95.4 | 99.8 | **100.0** | **100.0** | 85.7 | 96.4 | 95.4 | 99.8 | 85.0 | 99.5 | 94.7 | **99.4** |
| PiD ($M_{YUV}$, floor) | **100.0** | **100.0** | **100.0** | **100.0** | 97.5 | 99.4 | **97.6** | 99.8 | 97.6 | 99.5 | **100.0** | **100.0** | 84.2 | 91.2 | **97.6** | 99.6 | **97.4** | 99.9 | **96.9** | 98.8 |

the low-level representation-based method NPR by 9.4%, and the pretraining-based method C2P-CLIP by 2.2%.

**Evaluation on Self-Synthesis.** To comprehensively evaluate the performance of detection models over different GAN sources, we test models on Self-Synthesis with 9 advanced GAN models. All the models are trained on the training set of ForenSynths. Most previous methods cannot generalize well on this dataset. As in Table 4, our method surpasses the second-best NPR method (Tan et al., 2024d) by 1.2% on the average accuracy and by 2.8% on mAP.

### 4.3. Ablation Study

**Ablation on the transfomation matrix $M_t$.** We explore whether the performance of PiD is sensitive to the choice of transformation matrix $M_t$ on GenImage. Four matrix are used in this ablation as shown in Table 6. $M_{YUV}$, $M_{XYZ}$, and $M_{custom}$ map the pixel value from RGB color space to another color space. All these matrices has the same l1-norm $\|M_t\|_1 = 1$ and do not change the length of the

value range. $M_{diag}$ ($\|M_{diag}\|_1 < 1$) is a simple diagonal matrix and only scales the RGB value before the quantization. On the one hand, the results in Table 6 illustrate that even using the simple $M_{diag}$ can effectively improve the generalization performance of detection models. This indicates that the pixelwise decomposition can extract the essential noise information for AIGI detection. On the other hand, the choice of $M_t$ is still important to the detection performance, and $M_{YUV}$ achieves the best overall accuracy. See the Appendix for the detailed information on the matrix used in the ablation.

**Ablation on the scaling of matrix $M_t$.** The transformation matrix $M_t$ can affect the length of the value range of the mapped pixel vector, thus controlling the intensity $|R_{pid}(x)|$ of the residual representation. By scaling the $M_t$ ($M_{YUV}$ here) we can study the influence of $|R_{pid}(x)|$ on the performance. As shown in Figure 5, we measure the intensity of residual with MSE between the original pixel value and transformed pixel value. When $M_t$ is scaled with a factor larger than 1.0 (1.5 or 2.0), the average intensity of

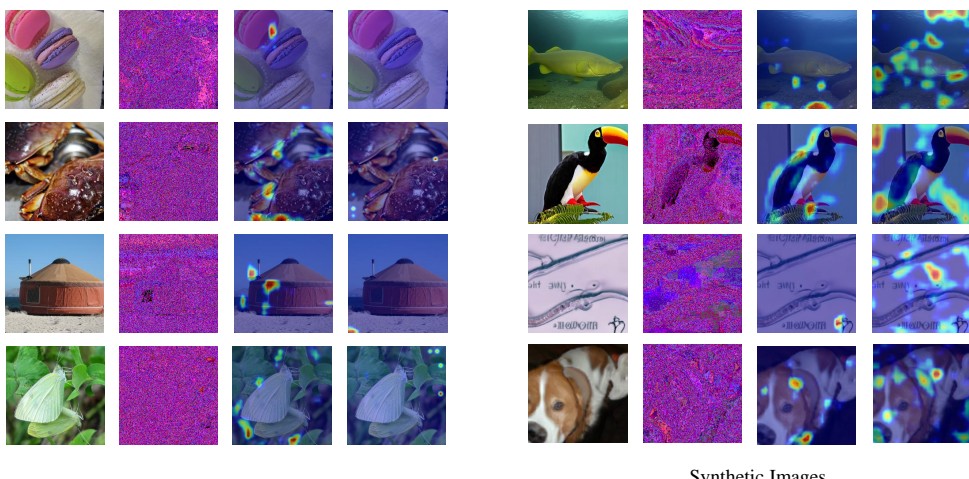

Real Images                    Synthetic Images

*Figure 4.* Visualization of GradCAM (Selvaraju et al., 2017) on GenImage. We visualize the GradCAM of the baseline RGB model (the 3rd column) and our residual model (the 4th column). The normalized residual information is also visualized in the 2nd column.

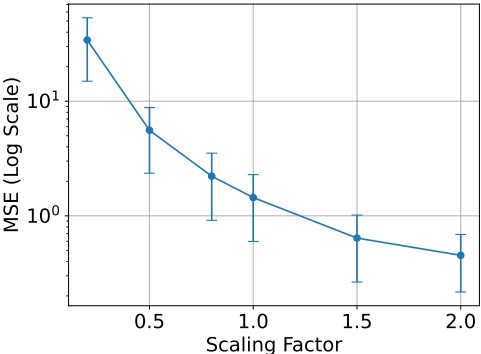

*Figure 5.* The pixel-level decomposition residuals vary in intensity based on different scaling factors. We compute the residual intensity over 10,000 randomly sampled pixel values. Log scale is applied for better visualization.

residual drops ($|R_{pid}(x)| < 1$) and causes a sparse residual representation. Evaluation results on GenImage in Table 5 illustrate that a large scaling factor (sparse residual representation) causes a large generalization performance drop. However, the generalization remains stable when using a scaling factor smaller than 1.0 from the results. Therefore, it is recommended to keep the l1-norm of $M_t$ smaller than 1.0 ($\|M_t\|_1 < 1$) in the application of PiD.

**Visualization of the GradCAM.** We visualize the attention of detection models with GradGAM (Selvaraju et al., 2017) in Figure 4. The residual of PiD is also visualized in the 2rd column. The synthetic images are generated from Midjourney (Mid, 2022), SDv1.4 (Rombach et al., 2022), ADM (Dhariwal et al., 2021) and BigGAN (Brock et al., 2018). The residual patterns between real and synthetic

*Table 5.* Ablation of the mapped value range on GenImage by scaling $M_t$. The matrix $M_t$ here is $M_{YUV}$, and the residual intensity is influenced by the scaling.

| Scaling factor | Midjourney | SDv1.4 | ADM | BigGAN |
|---|---|---|---|---|
| 0.2 | 93.6 | 99.2 | 87.4 | 91.2 |
| 0.5 | **96.7** | 98.7 | 92.8 | **97.9** |
| 0.8 | 92.6 | 98.8 | 92.0 | 91.3 |
| 1.0 | 95.5 | **99.6** | **96.0** | 96.2 |
| 1.5 | 84.6 | 93.1 | 69.0 | 79.6 |
| 2.0 | 79.6 | 93.3 | 63.1 | 76.4 |

*Table 6.* Ablation of the transformation matrix $M_t$ in PiD on GenImage. Four types of $M_t$ are used in the ablation, $M_{YUV}$: YUV color transformation, $M_{XYZ}$: XYZ color transfomration, $M_{diag}$: diagonal matrix, $M_{custom}$, customized full-rank matrix.

| Model | Midjourney | SDv1.4 | ADM | BigGAN |
|---|---|---|---|---|
| Base (RGB) | 67.5 | 98.7 | 55.9 | 61.6 |
| $M_{YUV}$ | 95.5 | **99.6** | 96.0 | **96.2** |
| $M_{XYZ}$ | **96.5** | 97.8 | **96.5** | 81.6 |
| $M_{diag}$ | 88.2 | 94.2 | 88.3 | 88.4 |
| $M_{custom}$ | 87.2 | 96.5 | 89.4 | 91.7 |

images exhibit some differences from the visualization. The residual of synthetic images are likely share the same pattern within a local region. From the attention map, the RGB model (the 3rd column) focus more on semantic object in the image. Compared with the baseline model, our residual model can distinguish the synthetic images from real images in the attention map. The results further illustrate the significance of the residual information in AIGI detection.

## 5. Conclusion

This paper has introduced PiD, an effective AIGI detection method that focuses on residual signals within images. By disentangling residual components from images, PiD uncovers underlying generation clues independent of semantic content, offering a computationally efficient and generalizable approach. Extensive cross-model experimental results have demonstrated the remarkable generalization performance of the proposed method. Future work could explore further optimization of the method and its application to a wider range of fake image detection scenarios. We hope that the findings in this work can also inspire the improvement of generative models.

## Acknowledgements

This work is supported in part by National Science Foundation for Distinguished Young Scholars under Grant 62225605, Project 12326608 supported by NSFC, "Pioneer" and "Leading Goose" R&D Program of Zhejiang (No.2025C02014), Ningbo Science and Technology Special Projects under Grant No. 2025Z028, and the Fundamental Research Funds for the Central Universities.

## Impact Statement

This paper presents work whose goal is to advance the field of Application-Driven Machine Learning. An AI-generated image detection method is studied and proposed in this paper. The proposed method has demonstrated potential in preventing the malicious applications of generative models and deepfakes, and may produce a positive social impact.

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

## A. Transformation Matrix

Details for the transformation matrix used in Table 6, $M_{YUV}$, $M_{XYZ}$, $M_{diag}$ and $M_{custom}$ are presented in this section. $M_{YUV}$, $M_{XYZ}$ and $M_{custom}$ are normal full-rank matrix. The l1-norm of these matrices are the same ($\|M_t\| = 1$). $M_{diag}$ is a full-rank diagonal matrix.

$$
M_{YUV} = \begin{bmatrix} 0.299 & 0.587 & 0.114 \\ -0.168736 & -0.331264 & 0.5 \\ 0.5 & -0.418688 & -0.081312 \end{bmatrix}, M_{YUV}^{-1} = \begin{bmatrix} 1.0 & 0.0 & 1.402 \\ 1.0 & -0.344136 & -0.714136 \\ 0.5 & -0.418688 & -0.081312 \end{bmatrix},
$$

$$
M_{XYZ} = \begin{bmatrix} 0.412453 & 0.357580 & 0.180423 \\ 0.212671 & 0.715160 & 0.072169 \\ 0.019334 & 0.119193 & 0.950227 \end{bmatrix}, M_{XYZ}^{-1} = \begin{bmatrix} 3.240479 & -1.537150 & -0.498535 \\ -0.969256 & 1.875991 & 0.041556 \\ 0.055648 & -0.204043 & 1.057311 \end{bmatrix},
$$

$$
M_{custom} = \begin{bmatrix} 0.06 & 0.63 & 0.27 \\ 0.3 & 0.04 & -0.35 \\ 0.34 & -0.6 & 0.17 \end{bmatrix}, M_{custom}^{-1} = \begin{bmatrix} 1.1844 & 1.5685 & 1.3482 \\ 0.9909 & 0.4756 & -0.5945 \\ 1.1284 & -1.4583 & 1.0876 \end{bmatrix},
$$

$$
M_{diag} = \begin{bmatrix} 0.412453 & 0.0 & 0.0 \\ 0.0 & 0.715160 & 0.0 \\ 0.0 & 0.0 & 0.950227 \end{bmatrix}, M_{diag}^{-1} = \begin{bmatrix} 2.424519 & 0.0 & 0.0 \\ 0.0 & 1.398288 & 0.0 \\ 0.0 & 0.0 & 1.05238 \end{bmatrix}.
$$

## B. Cross-Model Evaluation

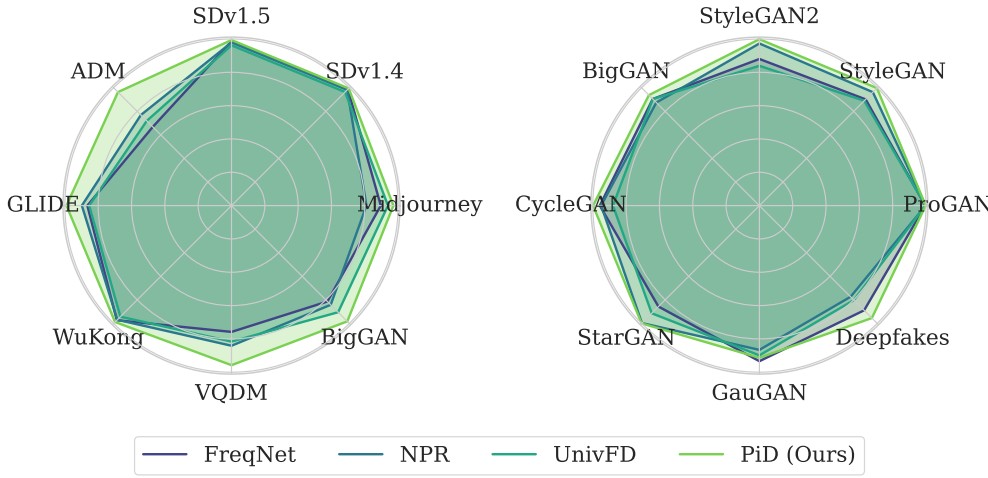

*Figure 6.* Cross-model accuracy (Acc) performance on GenImage (left) and ForenSynths (right) datasets.

### B.1. Comparison with Frequency and Pretrained Models

We visualize the evaluation results in Figure 6. Compared with other low-level residual-based methods like FreqNet (Tan et al., 2024c) and NPR (Tan et al., 2024d), our method achieves better generalization performance on some generative models like ADM (Dhariwal et al., 2021), VQDM (Egiazarian et al., 2024), BigGAN (Brock et al., 2018), and Deepfakes (Rossler et al., 2019b). The residual information used in PiD is not limited to high-frequency information only and can generalize to more synthesis methods. Compared to the pretraining-based UnivFD with a large ViT backbone, our model also achieves a better overall detection performance. The results illustrate the effectiveness and efficiency of our method.

### B.2. Comparison with LARE$^2$ on Multi-Source Training Set

To validate whether the performance is sensitive to the generative models used during training, we test models trained on different training sets on GenImage. As shown in Figure 7, we compare our model with LARE$^2$ (Luo et al., 2024),

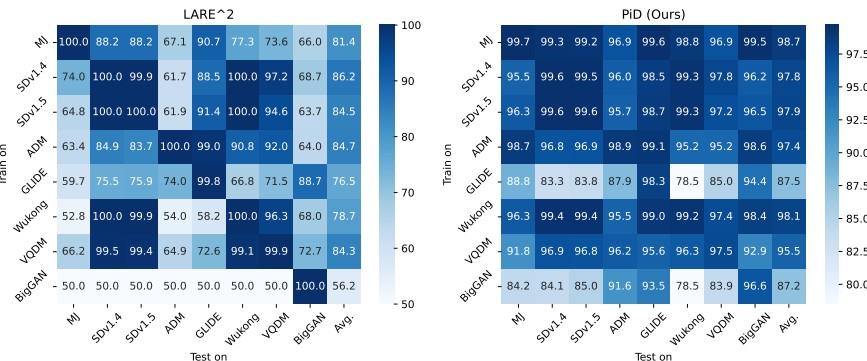

*Figure 7.* Cross-model evaluation with different training sets on GenImage. We compare the performance with LARE$^2$ with different training sets.

which is a reconstruction residual-based method in the latent space. The performance of LARE$^2$ is great when training on the SD model, but it cannot generalize well when training on other DM or GAN models like GLIDE or BigGAN. Since the reconstruction method usually relies on specific models like SD to extract residuals, the model bias may hinder the generalization of these methods. Our method does not require a generative model to extract residuals and achieves great generalization performance with different training sets as shown in Figure 7.

## C. Visualization

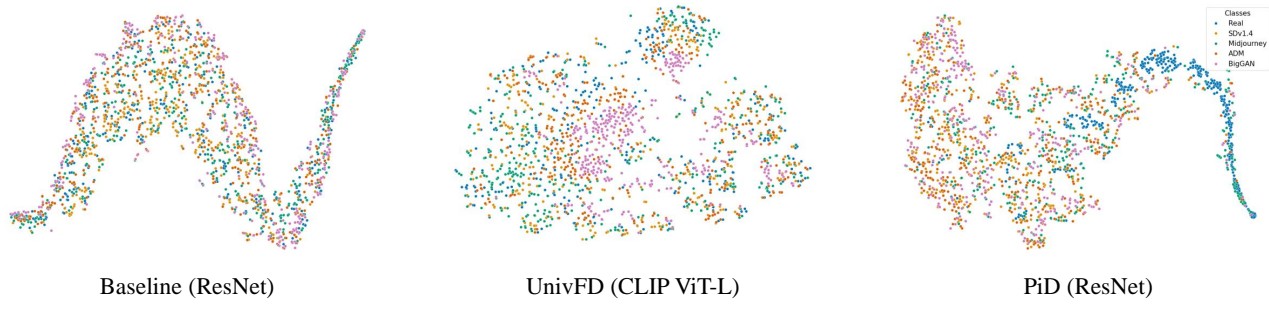

Baseline (ResNet)     UnivFD (CLIP ViT-L)     PiD (ResNet)

*Figure 8.* The t-SNE visualization of RGB (baseline and UnivFD) and residual (PiD) detection models on GenImage. With RGB images as input, real and synthetic images cannot be easily separated in the feature space. The residual information is less influenced by the visual content information. The real images (blue points) are distinguished from synthetic images in the feature space of residual models.

**Visualization of the feature space.** To explore the effectiveness of residual representation, we visualize the feature space of RGB and residual detectors with t-SNE (Van der Maaten & Hinton, 2008) on GenImage. As shown in Figure 8, with RGB input, real and synthetic images cannot be simply separated in the feature space of the baseline model. For UnivFD, which utilizes a frozen ViT-L (CLIP) backbone and generalizes well on some generative models, some images are clustered in the feature space. However, real images are still not clustered and cannot be distinguished. The residual representation in PiD is less influenced by the visual semantic information. Real images are clustered in the feature space of the residual detector and separated from synthetic images. The results further demonstrate the effectiveness of our method in AIGI detection.

## D. Computational Cost

Our method shows an advantage in the computational cost compared with previous methods, as shown in Table 7. First, our method does not increase the learnable parameter numbers of the detector and requires a light weight CNN network only. The overall number of parameters (#Params.) and operations (GFLOPs) is much lower than some methods that rely on pretrained networks like UnviFD (Ojha et al., 2023). Second, the computational cost of the proposed pixelwise

| Model | #Params. | GFLOPs | Inference Time (ms) |
|---|---|---|---|
| Baseline (RGB) | 1.4M | 1.73 | 29.13 |
| FreqNet | 1.8M | 2.28 | 142.00 |
| DIRE | 23.5M | 4.09 | 2425.44 |
| UnivFD | 85.8M | 17.58 | 63.79 |
| Ours | 1.4M | 1.73 | 32.92 |

*Table 7.* Comparison of different methods on the computational cost. We compare different models and report the parameters numbers and inference time of each model on the same device.

decomposition operation is also more efficient than the frequency transform or reconstruction process (like DDIM in DIRE). From Table 7, the inference time (a single forward pass) of our method is similar to that of the baseline model without extra operations (around 30 ms), while frequency-based method FreqNet (142.00 ms) and reconstruction-based DIRE (2425.44 ms) are much slower than ours and the baseline model.

