# OpenReview forum: "PiD: Generalized AI-Generated Images Detection with Pixelwise Decomposition Residuals"
_ICML.cc/2025/Conference — ICML 2025 poster_

### Official Review · Reviewer_4ooD · 2025-03-03

**Overall Recommendation:** 2

**Summary:**

In this paper, the authors propose extracting the “residual” of images to detect AIGC images. Specifically, the residual refers to artifacts introduced at the low-level visual features due to the generative model’s excessive focus on semantic content during the image generation process. These artifacts serve as distinguishing cues between AI-generated and real images. Extensive experiments demonstrate the effectiveness of this approach.

**Claims And Evidence:**

[+] Using image residuals as evidence for detection is clear and well-supported.

[-] However, what exactly the “residual” represents in an image, why it can be easily extracted in the YUV color space, what kind of information previous residual extraction methods have captured, and how the residual in this work differs from them all remain unproven.

**Essential References Not Discussed:**

N/A

**Experimental Designs Or Analyses:**

The experimental results demonstrate the effectiveness of the proposed model. However, it is recommended that the authors extend their method to facial synthesis datasets rather than focusing solely on general scene datasets.

**Methods And Evaluation Criteria:**

The proposed method, its implementation, and the chosen baselines are comprehensive. However, the authors are encouraged to include additional datasets, such as traditional face-swapping datasets, to investigate whether the extracted residuals remain effective for facial images.

**Other Comments Or Suggestions:**

A more thorough theoretical analysis of residuals is necessary, and additional experiments are also encouraged to further validate the approach.

**Other Strengths And Weaknesses:**

[-] The primary concern of the reviewers remains the precise definition of “residuals.” Why do these residuals inherently capture the artifacts unavoidably introduced during the generation process? What fundamental theoretical differences exist between residuals in real and fake images?

[-] Are there alternative residual processing methods beyond JPEG compression and the YUV color space? Is residual extraction possible for any model with an encoder-decoder structure? How can we ensure that these methods do not introduce additional biases, as seen in DIRE?

**Questions For Authors:**

Please see weaknesses.

**Relation To Broader Scientific Literature:**

The proposed method may offer insights for generalized deepfake detection by further exploring the distinctions between real and fake images.

**Theoretical Claims:**

The reviewer has checked the definition of the noise-aware residual representation R(x), and confirmed its feasibility for AIGC detection.

---

> ### Author Rebuttal · Authors · 2025-04-01
>
> We sincerely thank Reviewer 4ooD for the thoughtful comments. The responses to the questions are as follows.
>
> > *Q1: Why do these residuals inherently capture the artifacts?*
>
> **A:** Thanks.
> - Generally, the residual for an image input $x$ has the form $R(x) = x - \Phi(x)$ as described in the paper. Ideally, $ \Phi(x)$ represents the information that contributes most to the visual quality.
> - The training of generative models mainly focuses on content consistency, while the noise information is not well-modeled (as shown in Eq. 3). This information does not affect the overall visual quality of images, but can also reflect the difference between real and fake images.
> - Differences in high-frequency residual distribution are observed in [1]. While the distribution of other residuals cannot be well visualized, the test results demonstrate that residuals like DIRE or the proposed PiD also exhibit discrimination capability in AIGC detection.
>
> > *Q2: Other residual processing methods.*
>
> **A:** Some methods in previous work can also be classified as residual-based methods like high-frequency components (DCT/DWT/FFT) or reconstruction error (diffusion models). An encoder-decoder structure exists in these methods. However, learnable structures may be avoided to alleviate the bias in the residual signals. A simple operation that filters the content information shows advantages in generalization, as the proposed method.
>
> > *Q3: Results on facial synthesis datasets.*
>
> **A:** Thanks. The generalization of different source images is important. Some test sets in experiments are facial synthesis datasets, in which both real and fake images are faces, like AttGAN, STGAN, and Deepfakes. Our method can generalize well on these facial synthesis datasets at testing. To further explore the capability of the proposed method, we also test our method on high-quality facial synthesis data generated from HeyGEN (VFHQ as the real dataset), the accuracy is 86.95\% (much higher than the baseline 62.88\%), which shows the potential in broader application.
>
> [1] Frank J, Eisenhofer T, Schönherr L, et al. Leveraging frequency analysis for deep fake image recognition[C]//International conference on machine learning. PMLR, 2020: 3247-3258.

---

### Official Review · Reviewer_45ju · 2025-03-12

**Overall Recommendation:** 1

**Summary:**

This paper proposes Pixelwise Decomposition Residuals (PiD) to distinguish real images from synthetic ones. Based on the hypothesis that generative algorithms often overlook low-level signals, the authors decompose synthetic images in the RGB domain. Specifically, they convert images to the YUV color space, apply quantization, revert them back to RGB, and compute residuals. These residuals are then used as input to train a neural network detector. The method claims generalization capabilities across three datasets.

**Claims And Evidence:**

- The abstract and Section 1 repeatedly emphasize that existing methods are computationally complex, while the proposed approach is more efficient. However, no comparative or quantitative evaluation of computational complexity is provided in the methodology or experiments. Thus, the claimed contribution of "a computationally efficient method" lacks empirical support.

- Similarly, the abstract and Section 1 argue that prior work tends to overfit to generator-specific artifacts, advocating a generator-free design. Yet, the experiments follow the same training paradigm as existing methods by using data from fixed generators (e.g., ProGAN or SDv1.4). This contradicts the claimed novelty of avoiding generator-specific biases.

**Essential References Not Discussed:**

D. Cozzolino, et al. "Zero-Shot Detection of AI-Generated Images", in ECCV'24.

**Experimental Designs Or Analyses:**

- Inconsistent Benchmarking: The results in Tables 1-3 are copied from different existing works, rather than replicated by the authors themselves. There is a serious deviation in the results caused by inconsistent raw test data. For example, the original papers of DRCT and C2P-CRIP both tested the results of UnivFD on the GenImage dataset, which showed an average Acc of 79.45 and 88.8, respectively. However, the authors clearly copied the results of the latter (C2P-CRIP) directly and entered them into Table 3. So, why not copy the results of DRCT?

- Cherry-Picked Variants: Many existing works have different variants, for example, CNNSpot has two data augmentation versions with probabilities of 0.1 and 0.5, while UnivFD has two implementation methods based on Nearest Neighbor and Linear Probe. However, the authors are very **tricky**. Table 2 only shows the results of its poor variants for UnivFD. For example, the AP of UnivFD in DALLE detection in Table 2 is 88.45, while the original result of UnivFD shows that it can achieve an AP of 97.15 on DALLE in the variant of Linear Probe.

**Methods And Evaluation Criteria:**

Most of the existing reconstruction-based discrimination work requires the introduction of external knowledge (such as DDIM adopted by DIRE), but the author believes that directly "separating" a residual noise from the image can achieve detection with strong generalization. My question is:
- Essentially, these noises also exist in RGB images and are inputted into the neural network for training. So, why can't end-to-end training automatically learn the representation corresponding to these highly generalized noises from RGB images when the loss and final goal are consistent?
- If this residual noise already has strong discriminability, then simpler networks (such as simple MLP or even a single layer of linear probe) should be used in the future to achieve good discriminability. However, the author did not provide sufficient support in this regard.

**Other Comments Or Suggestions:**

Line 184: Replace "M\_high" with "M\_{high}" for proper LaTeX formatting.
Avoid using "raw" to describe original RGB images, as it may confuse readers with the RAW image format.

**Other Strengths And Weaknesses:**

NA

**Questions For Authors:**

Why is there only one variant of PiD proposed in Table 4, instead of two as shown in Tables 1-3?

**Relation To Broader Scientific Literature:**

NA

**Theoretical Claims:**

While Figure 3 attempts to justify the use of color conversion for decomposition, critical questions remain unaddressed:
- How does the method handle grayscale or monochrome images?
- Why is the transformation matrix M_t designed as 3×3? Why are quantization functions (e.g., round or floor) chosen without ablation studies?
- Would decomposition based on camera-specific artifacts (e.g., Noiseprint [TIFS’19] or Noiseprint++ [CVPR’23]) yield better results?

---

> ### Author Rebuttal · Authors · 2025-04-01
>
> We sincerely thank Reviewer 45ju for the detailed comments. The responses to the questions are as follows.
>
> > *Q1: Computational efficiency.*
>
> **A:** Thanks. We compare the computation cost of our method and previous methods. The inference time is only slightly slower than the baseline model, while it is **significantly faster** than other residual-based methods or pretraining-based methods like UnivFD.
>
> |Models|#Params|GFlops|Time (ms)
> |-|-|-|-
> |Baseline|1.4M|1.73|29.13
> |FreqNet|1.8M|2.28|142.00
> |DIRE|23.51M|4.09|2425.44
> |UnivFD|85.8M|17.58|63.79
> |Ours|1.4M|1.73|32.92
>
> > *Q2: Questions on the training paradigm.*
>
> **A:** Thanks. We emphasize the generalization capability of methods rather than the training paradigm. Therefore, to have a fair comparison with previous work, we follow the same setting commonly used in AIGC detection (training on seen models and testing on both seen and unseen models). Improvement has been demonstrated in the experiments and meet our claim.
>
> > *Q3: Questions on end-to-end training and networks.*
>
> **A:** Thanks. We clarify the training process as follows.
> - 1) Without guidance, networks are **not guaranteed to** focus on the residual information to classify the data in the training set. The decomposition filters out non-residual information and forces the model to leverage the residual information in classification.
> - 2) Residual input is still in an image form, therefore using a convolutional network to capture the patterns is a natural choice as in previous work like FreqNet/DIRE/NPR. The results fairly reflect the capability of different inputs.
>
> > *Q4: Module settings (image channels, transformation matrix, and quantization).*
>
> **A:** Thanks. We explain the questions on the experiment settings as follows.
> - 1) For grayscale or monochrome images, we *expand* the channels from 1 to 3 to achieve consistent channel numbers during training.
> - 2) A full-rank $3\times3$ transform matrix is *simple and invertible*. It can map an image forward to a different color space and back to the RGB space to compute the residual information. This is commonly used in color transformation as a basic setting.
> - 3) Rounding and truncation (floor) are two main *basic and standard* quantization operations in signal processing and computation [1]. Using standard quantization functions is computationally efficient and demonstrates the effectiveness of the framework. We would like to clarify these points in the manuscript.
>
> > *Q5: Comparison with camera-specific artifacts.*
>
> **A:** NoisePrint++ is a kind of image representation for image anti-spoofing extracted with networks, **rather than an explicit decomposition** of image information. We compare the NoisePrint++ as input with our method using the same backbone networks. The accuracy (%) results are as follows. On some test sets, it shows improvement on some test sets but the overall generalization is still limited in AIGC detection. The leverage of learnable image representation in AIGC detection requires further study.
>
> |Models|MidJourney|SDv1.4|SDv1.5|ADM|GLIDE|Wukong|VQDM|BigGAN|Avg.
> |-|-|-|-|-|-|-|-|-|-
> |Baseline|67.46|98.68|98.68|55.90|62.80|97.93|49.52|61.62|74.07
> |NoisePrint++|77.18|96.92|96.74|62.10|57.37|92.47|89.61|47.79|77.52
> |Ours|97.16|99.48|99.36|96.34|99.04|98.82|95.76|98.26|98.03
>
> > *Q6: Questions on the benchmarking.*
>
> **A:** Thanks for the suggestion.
> - 1) We intend to compare our method with the best-reported results under the same test setting. Therefore, we cite the higher UnivFD result of 88.8 here. However, the results in DRCT are convincing, and we have reproduced the results of UnivFD with an accuracy of 80.7. This result can be included in the table for future reference.
> - 2) The results of UnivFD are not deliberately chosen in Tables 1 and 2. Since we follow the SOTA results from C2P-CLIP in Tables 1 and 2, some of the reported results of early methods like UnivFD are cited as slightly different from the original paper. We would like to cite the best-reported results under the same setting and check them accordingly.
>
> > *Q7: Suggestions on the reference, writing, and presentation.*
>
> **A:** We have followed your suggestion and modified the manuscript accordingly. The citation is added and discussed in the related works.
>
> > *Q8: Results on Self-Synthesis.*
>
> **A:** Since the results using round are slightly better on GenImage and UniFakeDetect, we use round as the quantization when testing on Self-Synthesis. The results (Acc/AP) of the floor are as follows, and **the performance is still similar**.
>
> |Models|AttGAN|BEGAN|CramerGAN|InfoMaxGAN|MMDGAN|RelGAN|S3GAN|SNGAN|STGAN|Avg.
> |-|-|-|-|-|-|-|-|-|-|-
> |Ours (round)|100.0/100.0|99.9/100.0|95.4/99.7|95.4/99.8|95.4/99.8|100.0/100.0|85.7/96.4|95.4/99.8|85.0/99.5|94.7/99.4
> |Ours (floor)|100.0/100.0|99.9/100.0|97.5/99.4|97.6/99.8|97.6/99.8|100.0/100.0|84.2/91.2|97.6/99.6|97.4/99.9|96.8/98.8
>
> [1] Quantization. IEEE transactions on information theory.

---

### Official Review · Reviewer_uAyf · 2025-03-13

**Overall Recommendation:** 3

**Summary:**

In this paper, a new discriminant method PID is proposed to detect whether an image is generated by a generative model. This paper explores the impact of noise residual distribution on the discriminant model and believes that the noise space of the image can be represented by color space transformation. By training the transformed residual information, a discriminant model that far exceeds RGB images and does not rely on semantic features can be obtained. The model achieves SOTA level performance and shows good generalization performance.

**Claims And Evidence:**

In this paper, the claims are sufficient and supported through empirical experimental results as follows:
1. The PID method mentioned in the paper uses quantization operation after color space transformation to extract residual noise. The extraction costs little overhead and is efficient to generalize in detection tasks.
2. The paper claims that the model has strong generalization capabilities and is independent of semantic information. The residual extraction process does not rely on specific generative models and gets rid of the semantic disruption.

**Essential References Not Discussed:**

The provided related works are sufficient to understand the key contributions of the paper.

**Experimental Designs Or Analyses:**

The paper mentions a large number of experiments, including comparative experiments compared with previous methods, test experiments on multiple datasets, and ablation experiments. The overall experimental content is detailed.

**Methods And Evaluation Criteria:**

The proposed method utilizes color space transformation and quantization to extract residual image which does not rely on specific generative models and guarantees the generalizability. The proposed method is evaluated on open-source datasets which are widely used in such researches. The accuracy and average precision criteria used in this paper verifies the efficiency of the method.

**Other Comments Or Suggestions:**

No more comments.

**Other Strengths And Weaknesses:**

Strengths:
1. This article is highly innovative, and it obtains the idea of extracting noise space from the JPEG image compression method.
2. Compared with other methods, this method has lower computational complexity and provides a new perspective for solving this problem.

Weaknesses:
1. The variety of real image datasets is lacking when verifying the generalizability of the proposed method. The experiments should include more datasets of different distribution.
2. There should be more qualitative results showing the differences of residual noise in different datasets. The GradCAM is not sufficient to verify the motivation of the proposed method.
3. In Figure.7, the evaluation results have a huge drop when training on GLIDE and BigGAN which means that the classifier still has bias on specific generative fingerprints. The causes of such results should be further discussed.

**Questions For Authors:**

No further questions.

**Relation To Broader Scientific Literature:**

Current methodologies address this challenge through two complementary lenses: low-level artifact analysis and high-level semantic cues. Low-level methods target severe statistical anomalies caused by the content generation process: CNN-Spot employs data augmentation to improve generalization, while BiHPF amplifies alpha shadows through dual high-pass filters, LGrad implements gradient-based patterns, NPR models pixel relationships, and random mapping features reveal form-specific distortions. In contrast, high-level methods exploit annotation inconsistencies in synthetic content: UnivFD employs CLIP embedding for zero-shot detection, FatFormer combines frequency analysis with language alignment from CLIP, and LASTED utilizes text-guided contrastive learning to identify mismatches between visual and textual semantics, combining low-level artifact detection with high-level semantic reasoning.

**Theoretical Claims:**

In this paper, the core argument is that the residual between the image after color space transformation and the original image contains the noise space information in the image generation process. However, it is not clear enough that the color space transformation can correspond to more complete color space information. This spatial change is beneficial for predicting the noise distribution of the image, but the completeness of using the spatial variation residual to fully represent the noise distribution is still lacking in the paper.

---

> ### Author Rebuttal · Authors · 2025-04-01
>
> We sincerely thank Reviewer uAyf for the constructive comments, and the responses are as follows.
>
> > *Q1: Include real image datasets of different distributions.*
>
> **A:** Thanks. The real image distribution is an important factor during testing. The test datasets UniversalFakeDetect and Self-Synthesis used in the experiments are **composed of multi-source test sets.** The test real images are from COCO, ImageNet, LAION, LSUN, CelebA, and FF++, and the training set used from ForenSynths is single-source (only 4-class LSUN images). Therefore, the test setting has met the requirement of diversity. We will make it clearer as in previous work.
>
> > *Q2: More qualitative results.*
>
> **A:** Thanks for the suggestion. We use GradCAM and residual images to illustrate that the model trained with the residual signal **attends to more AIGC artifacts correctly** than the baseline model. The activation of the baseline model falsely appears on the real images with the RGB input.
> We would like to provide the visualization of different residuals (frequency or reconstruction) on different generative models to compare them in the manuscript or supplementary material.
>
> > *Q3: Performance on GLIDE and BigGAN.*
>
> **A:** Thanks. Although the training method is crucial to the generalization performance, the training data distribution is still important. There might be two main reasons why using GLIDE and BigGAN as the training set harms the generalization.
> - First, the data of GLIDE and BigGAN in GenImage has a **smaller image size** than other classes. The resolution of BigGAN is the smallest ($128\times 128$) while the resolution of real images is closer to $512\times 512$. Directly training on them may cause a bias corresponding to the image size.
> - Second, the **image quality** of GLIDE and BigGAN is visually worse than other generative models in GenImage. Images are blurred in GLIDE, and the object is unclear in BigGAN subsets. This causes a large gap in generative artifacts with other models.
>
> Though achieving great generalization performance is difficult in these two training sets, our method largely improves the overall performance and shows an advantage under hard settings.

---

> > ### Comment · Reviewer_uAyf · 2025-04-09
> >
> > We appreciate the authors' detailed responses and revisions, which have adequately addressed our concerns. The additional clarifications on dataset diversity, qualitative results (GradCAM/residual visualizations), and performance analysis on GLIDE/BigGAN have strengthened the manuscript. We maintain our prior recommendation and thank the authors for their efforts.

---

### Official Review · Reviewer_jvUK · 2025-03-14

**Overall Recommendation:** 3

**Summary:**

This paper focuses on Generalized AI-generated image detection via learning low-level signals (residual components) from image compression. To achieve this, the authors map the pixel vector to another color space (e.g., YUV), quantize the vector, and map back to the RGB space. Afterward, the quantization loss is taken as the above low-level signals for training a model to detect AI-generated images. The advantage over the existing works lies in an easily implemented pipeline without introducing cumbersome generative models or Large Language Models.

**Claims And Evidence:**

The main intuition starts from how to find a computationally simple and universal forgery artifact without relying on generator-specific cues. However, this work lacks an analysis of why YUV or other transformation matrics are good at discovering these forgery artifacts.

**Essential References Not Discussed:**

No

**Experimental Designs Or Analyses:**

It seems that  YUV is the best choice, showing a big advantage over other transformation matrics. Considering this,
a deep theoretical analysis of YUV is best will enhance this paper greatly. Currently, the theoretical contribution is a little weak.

**Methods And Evaluation Criteria:**

The authors provide an extensive evaluation of 3 widely used datasets with 26 generative models. The overall evaluation criteria make sense to me.

**Other Comments Or Suggestions:**

1. Provide more experiment details of ablation studies.
2. I don't see much difference between Image Compression Residuals and Image reconstruction Residuals in the existing works. The authors are suggested to add theoretical analysis to distinguish two aspects.

Overall, my current rating is borderline.

**Other Strengths And Weaknesses:**

Strengths
1. The overall idea is simple and easy to follow.
2. The proposed methods do not rely on cumbersome VLM and still achieve SOTA results.
3. The authors provide an extensive evaluation of 3 widely used datasets with 26 generative models. The overall evaluation criteria make sense to me.

Weaknesses
1. Lack the inference speed information for the proof of its computational effectiveness.
2. Lack the theoretical analysis of why the AI-generated images are more likely to fail in the quantization process.
3. This work lacks an analysis of why YUV or other transformation matrics are good at discovering these forgery artifacts.
4. The pixel-wise decomposition residual is a well-known concept. A theoretical analysis could strengthen this paper.

**Questions For Authors:**

1. In the ablation study, what dataset is used in Table 6?
2. It seems that  YUV is the best choice, showing a big advantage over other transformation matrics. Considering this,
a deep theoretical analysis of YUV is best will enhance this paper greatly. Currently, the theoretical contribution is a little weak.

**Relation To Broader Scientific Literature:**

Inspired by compression algorithms, the authors reveal that the pixel-wise decomposition residual can benefit AI-generated image detection. However, the theoretical analysis is weak and lacks more discussions.

**Theoretical Claims:**

No theoretical claims.

---

> ### Author Rebuttal · Authors · 2025-04-01
>
> We sincerely thank Reviewer jvUK for the acknowledgement and constructive feedback. The response is as follows.
>
> > *Q1: Computation efficiency.*
>
> **A:** Thanks for the advice. To prove the computation efficiency of the proposed method, we compute the inference time (single forward pass) and the size of detector networks for different methods as follows. The transformation introduced in our method only adds a little overhead to the baseline cost. Compared with previous methods using reconstruction and frequency operation, the transformation is efficient. The model is also lightweight compared with methods relying on VLM like UnivFD or C2P-CLIP.
>
> |Models|#Params|GFlops|Time (ms)
> |-|-|-|-
> |Baseline|1.4M|1.73|29.13
> |FreqNet|1.8M|2.28|142.00
> |DIRE|23.51M|4.09|2425.44
> |UnivFD|85.8M|17.58|63.79
> |Ours|1.4M|1.73|32.92
>
> > *Q2: Dataset used in Table 6.*
>
> **A:** Thanks. Datasets used in Table 6 are subsets of GenImage, like in Table 5. We will clarify it in the manuscript.
>
> > *Q3: Difference between Image Compression Residuals and Image Reconstruction Residuals.*
>
> **A:** Thanks. The difference between these two concepts is as follows.
> -    Image Reconstruction Residuals used in the paper specifically describe the reconstruction error of generative models used in previous works like DIRE.
> -    The concept of Image Compression Residuals is more general in this paper. The compression operation requires an encoder-decoder structure that is not necessarily a neural network (e.g., JPEG compression or PiD). The bias of the generative model may not appear in Image Compression Residuals. We will clarify the concepts in the manuscript.
>
> > *Q4: Analysis on the transformation.*
>
> **A:** Thanks. We find that the generative model mainly focuses on the content consistency and hypothesize that the overlooked noise information is discriminative in AIGC detection. To verify the hypothesis, we propose the PiD to extract the noise residual during detection.
>
> - *Noise contributes little to the training with original input*. Suppose the image input is $x = u + \epsilon$ (simplified as vector), considering the simple first linear transformation $f(x) = Wx$ in the network with the loss $L$. The gradient of parameter $W$ can be decomposed into two parts $\frac{\partial L}{\partial f}\frac{\partial f}{\partial W} = \frac{\partial L}{\partial f}x^T = \frac{\partial L}{\partial f}u^T+\frac{\partial L}{\partial f}\epsilon^T$. Given $|u| \gg|\epsilon|$, the gradient is dominated by the main component $u$.
>
> - *PiD filters out the main component while maintaining the noise information.* To verify the effectiveness of the noise part, we subtract the main component $u$ with the transformation and quantization techniques (noted as $T(\cdot)$). The residual $R(x) = x - T(x) = u + \epsilon - T(u)$. Given that $T(u) = u + \epsilon'$, $R(x) = \epsilon - \epsilon'$ is a proxy representation of image noise, where $\epsilon'$ is controlled by the transformation matrix. Taking $R(x)$ as input, the contribution of the noise part can be verified by the experiments.
>
> Since the YUV transform is integrated in the JPEG algorithm, the quantization noise seems a common noise source for real images. And the generative model may also have special noise patterns mixed with other noise sources. By applying the transform and quantization, the main components of images are filtered out, and the noise patterns can be learned.

---

### Official Review · Reviewer_6H4b · 2025-03-24

**Overall Recommendation:** 3

**Summary:**

This paper proposes a novel framework for detecting AI-generated images (AIGIs) based on pixel-wise image residuals. Residuals are extracted from the quantization error of color space transform and used to train a binary classification model. The results demonstrate promising performance in detecting AIGIs.

**Claims And Evidence:**

See **Other Strengths And Weaknesses** and **Questions For Authors**.

**Essential References Not Discussed:**

I don't observe any significant omissions of essential references.

**Experimental Designs Or Analyses:**

See **Other Strengths And Weaknesses** and **Questions For Authors**.

**Methods And Evaluation Criteria:**

The proposed method and evaluation criteria make sense for the problem and application.

**Other Comments Or Suggestions:**

S1:

For the data in Table 1-4, since the improvement of mACC/mAP/mean is already marginal, I suggest highlighting the performance comparison of each individual sub-dataset to facilitate the positioning of the performance on a specific sub-dataset.

S2:

The legends in Figure 3 overlap with the bars, which affects readability. I suggest repositioning the legend box to prevent this overlap. Additionally, renaming "w/o CVT error" and "w/o DCT error" to "DCT only" and "CVT only" would reduce cognitive load for readers.

S3:

Figure reference in line 713 is missing.

**Other Strengths And Weaknesses:**

**Strengths：**

S1:

The paper is well-motivated and easy to follow. The authors develop a novel understanding of quantization error in JPEG-based image compression as key information to detect AI-generated images.

S2:

The design of PiD is simple yet novel. In contrast to previous frequency-based methods, the PiD has advantages in accuracy.

S3:

Compared to reconstruction-based methods, PiD seems to be more computationally efficient, which makes it possible to be deloyed on a large scale.

S4:

The experiments are comprehensive and demonstrate the impressive performance of PiD.

**Weakness：**

W1:

The paper does not discuss the detection performance of compressed images, which may become a key limitation. Images in real-world scenarios, especially on social media, are generally compressed by JPEG (or other image compression algorithms). Since one of the important scenarios for AIGI detection is images from social media, it would be interesting to see whether this method is still effective after the image has been compressed to a certain extent, and lost valuable high-frequency information/color information. I would like to see the method’s performance on related datasets, e.g. VISION dataset [1].

W2:

Section 3.2 discusses the contribution of DCT and CVT in the detection pipeline. The results show that DCT contributes marginally and even reduces accuracy, which needs further discussion. FreqNet [2] uses FFT to train a similar CNN-based classifier and proves the high-frequency component extracted by FFT is also effective. It would be beneficial to perform comparative experiments between FFT and DCT, to better identify the importance of high-frequency components in detecting AI-generated images.

W3:

Although it seems that PiD will be faster than many reconstruction-based methods, comparative experiments on computational efficiency are still missing, and the scale of the model is also unknown. Given that existing models generally achieve fairly high accuracy on domain datasets, computational efficiency will become a key metric for new methods.

[1] Shullani, Dasara, et al. "VISION: a video and image dataset for source identification." EURASIP Journal on Information Security 2017.1 (2017): 1-16, https://doi.org/10.1186/s13635-017-0067-2.

[2] Tan, Chuangchuang, et al. "Frequency-aware deepfake detection: Improving generalizability through frequency space domain learning." *Proceedings of the AAAI Conference on Artificial Intelligence*. Vol. 38. No. 5. 2024.

**Questions For Authors:**

Q1:

In Table 6, the explanation of the baseline RGB is unclear. The paper does not seem to explain the pipeline of the baseline RGB method. Please consider adding relevant explanations.

Q2:

I am curious if this framework is still robust against adversarial attacks when the pipeline is white-box and known to the public. Since the downstream classification model (ResNet) is widely considered to be vulnerable to adversarial attacks [1], malicious attackers will be able to use gradient methods such as FGSM[1] or PGD[2] to generate human-imperceptible perturbations and add them back to the to AIGI images. It would be good to have some experiments to verify the robustness.

[1] Goodfellow, Ian J., Jonathon Shlens, and Christian Szegedy. "Explaining and harnessing adversarial examples." *arXiv preprint arXiv:1412.6572* (2014).

[2] Madry, Aleksander, et al. "Towards deep learning models resistant to adversarial attacks." *arXiv preprint arXiv:1706.06083* (2017).

**Relation To Broader Scientific Literature:**

See **Other Strengths And Weaknesses** and **Questions For Authors**.

**Theoretical Claims:**

The authors claim that it’s an application-driven machine learning paper. The paper doesn't make strong theoretical claims. It primarily demonstrates its empirical effectiveness.

---

> ### Author Rebuttal · Authors · 2025-04-01
>
> We sincerely thank Reviewer 6H4b for the thoughtful and constructive feedback. The response is as follows.
>
>
> > *Q1: Performance on related datasets.*
>
> **A:** Thanks for the advice. We conducted the perturbation experiments on the GenImage dataset (SDv1.4 as an in-domain test set and other models as out-of-domain test sets). After the perturbation of JPEG and Gaussian blur (GB), the results are as follows. The **performance drop (AP) is similar to the baseline model** with normal RGB input, while the **overall results remain higher** than the baseline model. We will test the model on the source identification dataset VISION.
>
> |Pert.|RGB (ID)|Ours (ID)|RGB (OOD)|Ours (OOD)
> |-|-|-|-|-|
> |None|99.93|99.79|76.08|97.53
> |JPEG75|99.97|99.97|75.36|96.83
> |GB(k=7)|99.74|98.11|69.17|92.58
>
>
> > *Q2: Comparison between FFT and DCT.*
>
> **A:** Thanks. Although the DCT part is not the main component in our method, we further ablate the performance with DCT/FFT low/high-frequency components as input on GenImage. The accuracy cannot fully reflect the importance of frequency bands in AIGC detection. From the AP results, high-frequency bands (both DCT and FFT) perform better than low-frequency bands, which meets the results from previous works like FreqNet. However, frequency-based residual inputs seem sensitive to the selection of bands and cannot generalize well on some test sets. Furthermore, JPEG quantizes both high-frequency and low-frequency components in compression to different degrees, and the residual used in the paper is not pure low or high frequencies.
>
> |Model|Acc. (Mean)|AP (Mean)|
> |-|-|-|
> |Baseline|74.03|79.74|
> |FFT-LF|70.25|76.42|
> |FFT-HF|68.96|84.32|
> |DCT-LF|73.50|88.36|
> |DCT-HF|73.89|94.86|
>
> > *Q3: Comparison between computation costs.*
>
> **A:** We compare the computation cost as follows. We only count the parameters and flops of the detector network for all methods with the *fvcore* package. The inference time for a single forward pass includes the time of processing the input. The reconstruction-based method, like DIRE, takes the longest due to the extra generation process. Frequency-based operations like FFT in FreqNet and large pre-trained CLIP models used in UnivFD and C2P-CLIP also hinder the reference speed to different degrees. Our method only adds a little overhead compared with the RGB baseline and is efficient in computation.
>
> |Models|#Params|GFlops|Time (ms)
> |-|-|-|-
> |Baseline|1.4M|1.73|29.13
> |FreqNet|1.8M|2.28|142.00
> |DIRE|23.51M|4.09|2425.44
> |UnivFD|85.8M|17.58|63.79
> |Ours|1.4M|1.73|32.92
>
>
> > *Q4: Baseline explaination.*
>
> **A:** The baseline model is a simple ResNet model used in NPR (smaller than ResNet18). Our method is also based on the same model. We will make it clearer in the new version.
>
> > *Q5: Robustness to adversarial attacks*
>
> **A:** The white-box attack is hard to defend without a specially designed strategy. However, the success defense rate has improved significantly compared with the baseline model. The accuracy of the baseline model drops to 0 under a PGD attack and to 0.43 under an FGSM attack. Our model has an accuracy of 0.51 and 0.46 under PGD and FGSM attacks. The robustness has been largely improved under the PGD attack. A randomly selected model ensemble or training with adversarial samples may further alleviate the risk of attacks.
>
> > *Q6: Other suggestions.*
>
> **A:** Thanks for the kind reminder. We have modified the manuscript accordingly following the suggestions for better presentation.

---

### Decision · Program_Chairs · 2025-05-01

**Decision:**

Accept (poster)

**Comment:**

The submission received one reject, one weak reject, and three weak accepts. The authors provided detailed responses, which, while not overturning the negative rating, convinced the Area Chair that the submission's strengths and contributions outweigh its weaknesses.

----
### Summary of the paper:

This paper proposes PiD, a novel and lightweight method for AI-generated image detection based on pixelwise decomposition residuals. The core insight is that generative models primarily optimize semantic content while neglecting fine-grained noise components—what the authors refer to as "residuals". The method involves mapping images to an alternative color space (e.g., YUV), quantizing the pixel vectors, reconstructing back to RGB, and using the resulting residuals to train a downstream binary classifier.

### Strengths:

1. Novelty in Residual Extraction:

The idea of extracting quantization-induced residuals after color space transformation is novel and computationally simple. It sidesteps the need for reconstruction from generative models (e.g., as done in DIRE) or heavy semantic embeddings (e.g., CLIP in UnivFD).

2. Strong Empirical Results:

Across three major benchmarks (GenImage, UniFakeDetect, and Self-Synthesis), the method achieves competitive or superior results in accuracy and AP, particularly in generalization across unseen generative models, which is a known challenge in AIGC detection.

3. Computational Efficiency:

The method uses a lightweight ResNet variant and introduces only marginal overhead compared to RGB baselines (e.g., \~32.9 ms per image vs. 29.1 ms), while being much faster than methods like DIRE (\~2.4s) or CLIP-based approaches. This is backed by FLOPs and parameter count comparisons.

4. Detailed Rebuttal:

The authors provided a strong and thoughtful rebuttal, including:

- JPEG robustness experiments.
- FFT vs. DCT ablation.
- Efficiency benchmarks.
- Robustness to adversarial attacks.
- Clarification on YUV rationale and residual extraction theory.

### Weaknesses:

1. Theoretical Underpinning Could Be Stronger:

Several reviewers (jvUK, 4ooD, 45ju) noted that the motivation and formal justification for why residuals extracted in YUV (or any color space) are more effective than other low-level signals could be deepened. The rebuttal partially addressed this, but the paper still lacks a rigorous theoretical treatment.

2. Benchmarking Inconsistencies Raised:

Reviewer 45ju highlighted issues with cherry-picked baseline numbers (e.g., reporting weaker UnivFD variants), and use of results across papers without re-implementation under consistent settings. While the authors explain their rationale, this does slightly reduce trust in the experimental comparison.

3. Residual Conceptual Ambiguity:

The term "residual" in this context overlaps with prior work (e.g., DIRE, FreqNet, NoisePrint++), and some reviewers felt the distinction wasn't clearly defined. The rebuttal helps delineate compression residuals vs. reconstruction residuals, but more emphasis could be placed on how PiD's residuals differ in signal characteristics and generalization behavior.

4. Reviewer Conduct Concerns:

Reviewer 45ju’s tone and accusations (e.g., describing authors as “tricky”) are unprofessional and inconsistent with ICML review standards. Despite some valid points in the critique, such language undermines the review’s objectivity.


### Rationale for final rating:

Despite some remaining concerns about theoretical grounding and benchmarking rigor, I find the overall contribution of this paper to be solidly positive, especially in terms of practical impact, generalizability, and computational efficiency. The approach is original and potentially impactful in real-world deployments of AIGC detectors, particularly under resource constraints. The authors also demonstrated scientific diligence in addressing nearly all reviewer concerns in a transparent and responsive manner.

Therefore, I recommend a weak accept, particularly if space permits. A strengthened revision could be encouraged to include:

- A more detailed theoretical analysis of the residual mechanism.
- A clarified comparison with prior "residual"-based methods.
- Ensuring reproducibility with code and consistent benchmarking.

best,
AC